# New class of symmetric starlike functions subordinate to the generating function of Gregory coefficients

**Mohammad Faisal Khan** [ID]1*, **Mohammed Abaoud**2,
**Naeem Ahmad**3, **Muqrin A. Almuqrin**4

**1** Department of Basic Sciences, College of Science, and Theoretical Studies, Saudi Electronic University, Riyadh, Kingdom of Saudi Arabia, **2** Department of Mathematics and Statistics, Imam Mohammad Ibn Saud Islamic University (IMSIU), Riyadh, Saudi Arabia, **3** Department of Civil Engineering, College of Engineering, Qassim University, Buraydah, Saudi Arabia, **4** Department of Mathematics, College of Science in Zulfi, Majmaah University, Al-Majmaah, Saudi Arabia

* f.khan@seu.edu.sa

**Data availability statement:** All relevant data are within the manuscript.

## Abstract

Function theory research has long struggled with the challenge of deriving sharp estimates for the coefficients of analytic and univalent functions. Researchers have advanced the field by developing and applying a variety of approaches to get these bounds. In the current paper, we apply the technique of subordination, we define the family of symmetric starlike functions which is related to generating function of Gregory coefficients. We provide sharp bounds for the problem concerning the coefficients of the family of symmetric starlike functions connected to the generating function of Gregory coefficients by utilizing the notion of functions with positive real component. These problems include first five sharp coefficient bounds and Fekete-Szego problem along with the Hankel determinant of order three. Additionally, we explore the optimal bounds (sharp bounds) for two important functions, the logarithmic function and the inverse function within the same class of symmetric starlike functions which is related to generating function of Gregory coefficients.

## 1. Introduction

Complex analysis, a fundamental pillar of modern mathematical science, has far-reaching implications across various academic and scientific disciplines. Within its scope, Geometric Function Theory (GFT) stands out as a fascinating branch that delves into the geometric properties of analytical functions. This field has emerged as a vital tool in applied mathematics, with significant applications in engineering, electronics, fluid dynamics, nonlinear integral systems, contemporary mathematical physics, and differential equation, further highlighting the profound impact of complex analysis on our understanding of the world.

The Bieberbach conjecture, a historic problem in GFT, has been a focal point of research for over a century. Formulated by Bieberbach [1] in 1916, the conjecture pertains to the bounds of coefficients for univalent functions, which are a fundamental class of injective functions in complex analysis. The conjecture specifically states that for a function $h$ in the

**Funding:** This work was supported and funded by the Deanship of Scientific Research at Imam Mohammad Ibn Saud Islamic University (IMSIU) (grant number IMSIU-DDRSP2501).

**Competing interests:** The authors declare no competing of interest.

class $S$, with a Taylor-Maclaurin series expansion of (1), the coefficients are bounded by $|a_n| \le n$, for all $n \ge 2$, a relationship that has far-reaching implications for the field. The set $S$, first explored by Koebe in 1907, is a suclass of the larger class $\mathcal{A}$ of analytic functions. Bieberbach's work in 1916 marked the beginning of progress on this conjecture, with subsequent breakthroughs by Lowner [2], Garabedian and Schiffer [3], Pederson and Schiffer [4], and Pederson [5], gradually extending the proof to cover the cases $n = 3, 4, 5$, and 6. The conjecture was ultimately resolved for all $n \ge 2$ in 1985 by de Branges [6], who employed hypergeometric functions to deliver a comprehensive solution to this long-standing problem in complex analysis. The application of coefficient bounds to medical imaging data can improve the precision of diagnostic modalities such as MRI and CT scans, ultimately leading to the development of more targeted and effective treatments for a variety of health conditions.

Let $U$ denote the unit disk, defined as $U = \{t : t \in \mathbb{C} \text{ and } |t| < 1\}$, and let $\mathcal{A}$ denote a class of analytic functions satisfying the normalization conditions

$$h(0) = h'(0) - 1 = 0.$$

For each $h \in \mathcal{A}$ can be written as:

$$h(t) = t + \sum_{n=2}^{\infty} a_n t^n. \tag{1}$$

A function $h$ is considered univalent in $U$ if it has a one-to-one correspondence between $U$ and its image, meaning that each point in the image has a unique inverse image in $U$. In other words, for all $t_1, t_2 \in U$, if $h(t_1) = h(t_2)$ then $t_1 = t_2$. Let $S$ is the notation used to represent the set of all functions that have the property of univalence.

The class $\mathcal{P}$, consisting of analytic functions $p$ that satisfy the normalization condition $p(0) = 1$ and have positive real part, i.e., $Re(p(t)) > 0$ for all $t$ in the domain $U$ with

$$p(t) = 1 + \sum_{n=1}^{\infty} c_n t^n. \tag{2}$$

As is widely recognized, functions of the form (1) have an analytic inverse $h^{-1}$ in the domain $|w| < 1/4$, as stated by Koebe's theorem. Specifically, if $h$ belongs to the class $S$, its inverse can be expanded as

$$h^{-1}(w) = w + A_2 w^2 + A_3 w^3 + \cdots, |w| < 1/4. \tag{3}$$

Lowner [2] demonstrated that for functions $h$ in $S$ with inverses of this form, the coefficients $A_n$ are bounded by the sharp estimate

$$|A_n| \le \frac{(2n)!}{n!(n+1)!}. \tag{4}$$

The Koebe function's inverse, $K(t) = \frac{t}{(1-t)^2}$ is known to produce the sharpest bounds for coefficients $|A_n|$ $(n = 2, 3, ...)$ in (4) across all members of the class $S$. This has led to considerable interest in exploring the behavior of inverse coefficients for functions $h$ in specific geometric subclasses of $S$, as defined in (3). Although multiple authors have offered alternative proofs of the inequality (4), Yang proof [7] stands out for its simplicity and

clarity. Using the fact that $h(h^{-1}(w)) = w$, it is straightforward to observe from Eq (3) that

$$A_2 = -a_2, \ A_3 = 2a_2^2 - a_3 \text{ and } A_4 = -5a_2^3 + 5a_2 a_3 - a_4. \tag{5}$$

During the period from 1916 to 1985, significant research efforts were focused on establishing estimates for the $n^{th}$ coefficient bounds of different subclasses within the family of class $\mathcal{S}$, such as starlike ($\mathcal{S}^*$), close-to-convex ($K$), convex ($C$) and other related families of functions. The following definitions apply to these families:

$$\mathcal{S}^* \ = \ \left\{ h \in \mathcal{A} : Re\left( \frac{th'(t)}{h(t)} \right) > 0, \ t \in U \right\},$$

$$C \ = \ \left\{ h \in \mathcal{A} : Re\left( \frac{\left(th'(t)\right)'}{h'(t)} \right) > 0, \ t \in U \right\},$$

$$\mathcal{K} \ = \ \left\{ h \in \mathcal{A} : Re\left( \frac{th'(t)}{g(t)} \right) > 0, \text{ and } g \in \mathcal{S}^*, \ t \in U \right\},$$

$$\mathcal{BT} \ = \ \left\{ h \in \mathcal{A} : Re\, h'(t) > 0, \ t \in U \right\}.$$

Suppose $u$ is Schwarz analytic function in $U$, with $u(0) = 0$ and $|u(t)| < 1$ for all $t$ in $U$. If $h(t)$ and $g(t)$ are analytic in $U$, and $h(t) = g(u(t))$ for all $t$ in $U$, then $h$ is subordinate to $g$, written as $h \prec g$. Furthermore, if $g$ is univalent in $U$ and $h(0) = g(0)$, then $h(U) \subseteq g(U)$.

Ma and Minda [8] replaced the function $\frac{1+t}{1-t}$ with a more general analytic function $\varphi$ which satisfies the specific conditions $\varphi(0) = 1$, $\varphi(0) > 0$, and $\varphi$ maps $U$ onto univalently a region starlike with respect to 1 and symmetric with respect to the real axis. They investigated a comprehensive and general class of functions that encompasses various prominent classes as specific instances, providing a unified framework that includes various special cases:

$$\mathcal{S}^*(\varphi) = \left\{ h \in \mathcal{A} : \frac{th'(t)}{h(t)} \prec \varphi(t) \right\}.$$

The mathematical community recognizes the functions in the class $\mathcal{S}^*(\varphi)$ as Ma-Minda starlike functions, a designation honoring the contributions of mathematicians Ma and Minda. This general class has given rise to a diverse array of subfamilies, which have been extensively studied in (see, [9,10]). For instance, Kumar et al. [11] have considered generated function of Bell numbers

$$\varphi(t) = e^{e^{t-1}} \quad (t \in \mathbb{U}).$$

In recent research, Mendiratta et al. [12] investigated the exponential function

$$\varphi(t) = e^t \quad (t \in \mathbb{U})$$

and Goel and Kumar [13] have extensively investigated the Sigmoid function

$$\varphi(t) = \frac{2}{1 + e^{-t}} \quad (t \in \mathbb{U}).$$

They obtained important results on its structural representation, inclusion properties, coefficient bounds, growth behavior, distortion estimates, subordination relationships, and radii constants, respectively. Furthermore, Deniz [14] addressed the sharp coefficient problem in 2021, focusing on the specific function $\varphi(t) = e^{t + \frac{\lambda}{2} t^2}$ $(t \in \mathbb{U} \ \lambda \geq 1)$, a generating function for generalized telephone numbers. Meanwhile, Murugusundaramoorthy et al. [15] investigated ⊠-bi-pseudo-starlike functions with respect to symmetric points associated with Telephone numbers.

Geometric function theory has long investigated the upper bound for coefficients, which offers valuable insights into function behavior. The second coefficient bound is particularly crucial, as it leads to growth and distortion theorems, and additionally, the coefficient problem connected to Hankel determinants is another significant area of study. A key mathematical object in this study is the Hankel determinant. This determinant $H_{q,n}(h)$, $n \in \mathbb{N}$ are defined by

$$H_{q,n}(h) = \begin{vmatrix} a_n & a_{n+1} & ... & a_{n+q-1} \\ a_{n+1} & a_{n+2} & ... & a_{n+q} \\ \vdots & \vdots & \vdots & \vdots \\ a_{n+q-1} & a_{n+q} & ... & a_{n+2q-1} \end{vmatrix}. \tag{6}$$

This concept was introduced by Pommerenke [16,17]. Significant research, including the works of [18,19], has focused on determining sharp bounds for the second-order Hankel determinant, notably $H_{2,1}(h) = a_3 - a_2^2$ and $H_{2,2}(h) = a_2 a_4 - a_3^2$, within various subfamilies of class $\mathcal{S}$. For a comprehensive understanding, see [20–22]. The third-order Hankel determinant, $H_{3,1}(h)$, poses a considerable challenge, especially in establishing sharp bounds. The determinant $H_{3,1}(h)$ is

$$H_{3,1}(h) = a_3(a_2 a_4 - a_2^3 3) - a_4(a_4 - a_2 a_3) + a_5(a_3 - a_2^2)$$

and has been extensively studied in [23–25]. Babalola [26] first investigated $H_{3,1}(h)$ for the $\mathcal{K}, \mathcal{S}^*$, and $\mathcal{BT}$ families. Zaprawa [27] later extended these findings in 2017, proposing non-sharp bounds:

$$|H_{3,1}(h)| \leq \begin{cases} \frac{49}{540} & \text{for } h \in C, \\ 1 & \text{for } h \in \mathcal{S}^*, \\ \frac{41}{60} & \text{for } h \in \mathcal{BT}. \end{cases}$$

Research efforts continued to enhance these bounds, specifically for the $\mathcal{S}^*$ class [28,29]. Ultimately, the sharp bounds for these determinants were established for the $C, \mathcal{S}^*$, and $\mathcal{BT}$ classes, as presented in [30–32]:

$$|H_{3,1}(h)| \leq \begin{cases} \frac{4}{135} & \text{for } h \in C, \\ \frac{4}{9} & \text{for } h \in \mathcal{S}^*, \\ \frac{1}{4} & \text{for } h \in \mathcal{BT}. \end{cases}$$

Further advancements by Barukab et al. [33] and Lecko et al. [34] have determined the sharp bounds for $|H_{3,1}(h)|$ for other classes:

$$\mathcal{S}^*\left(\frac{1}{2}\right) = \left\{ h \in \mathcal{A} : Re\left(\frac{th'(t)}{h(t)}\right) > \frac{1}{2}, \ t \in U \right\},$$

$$\mathcal{R}_s = \left\{ h \in \mathcal{A} : h'(t) \prec 1 + \sinh^{-1} t, \ t \in U \right\}.$$

Those seeking a deeper understanding of third-order Hankel determinants in recently discovered subfamilies of univalent functions are encouraged to explore the works cited in [35,36] for further details and analysis:

$$\left( \begin{array}{cccc} \text{Authors:} & \varphi(t) & \text{Sharp Bound} & \text{Reference} \\ & & & \\ \text{S.Bangaand and S.S.Kumar} & \sqrt{1+t} & \frac{1}{36} & [37] \\ \text{K. Ullah et al.} & 1 + \tanh(t) & \frac{1}{9} & [38] \\ \text{Shi et al.} & 1 + \sin t & \frac{1}{9} & [39] \\ \text{Riaz et al.} & \frac{2}{1+e^{-t}} & \frac{1}{36} & [40] \\ \text{B.Rath et al.} & \frac{1}{1-t} & \frac{1}{9} & [41] \\ \text{Z.G. Wang et al.} & 1 + \sinh t & \frac{1}{9} & [42] \end{array} \right).$$

A class $\mathcal{S}_{3l}^*$ of starlike functions related with three-leaf-shaped region, was introduced by Gandhi in [43] as:

$$\mathcal{S}_{3l}^* = \left\{ h \in \mathcal{A} : \frac{th'(t)}{h(t)} \prec 1 + \frac{4}{5}t + \frac{1}{5}t^2, \, t \in U \right\}.$$

Tang et al. [44] studied the class

$$\mathcal{S}_{3l,s}^* = \left\{ h \in \mathcal{A} : \frac{2th'(t)}{h(t) - h(-t)} \prec 1 + \frac{4}{5}t + \frac{1}{5}t^2, \, t \in U \right\}$$

of symmetric points with three-leaf-shaped region. In 2021, Mustafa and Murugusundaramoorthy [45] introduced Mocanu-type bi-starlike functions associated with shell-shaped regions, deriving coefficient bounds and Hankel determinants. Separately, Murugusundaramoorthy et al. [46] explored coefficient functionals for a class of bounded turning functions connected to the three-leaf function. Inspired by aforementioned investigation, we explore the function $\varphi$, which transforms $U$ into a starlike region centered at 1, with coefficients that are the Gregory coefficients. These coefficients, similar to Bernoulli numbers, are rational numbers that decrease in value $\frac{1}{2}, \frac{1}{12}, \frac{1}{24}, \frac{19}{720}, ...$ and are essential in various numerical analysis and number theory contexts. They were initially discovered by James Gregory in 1671 and have since been a subject of interest in mathematics. Gregory coefficients are notably among the most frequently rediscovered mathematical entities. These coefficients have been identified by various names, including reciprocal logarithmic numbers, the second kind of Bernoulli numbers, Cauchy numbers, and others. Our focus in this paper is on the generating function $G_n$ of Gregory coefficients, as described in (see [47,48]), which is defined as follows:

$$\varphi(t) = \frac{t}{\ln(1+t)} = \sum_{n=0}^{\infty} g_n t^n \quad (|t| < 1). \tag{7}$$

Clearly, $g_n$ for $n = 0, 1, 2, 3, 4, 5$ are

$$g_0 = 1, g_1 = \frac{1}{2}, \, g_2 = -\frac{1}{12}, \, g_3 = \frac{1}{24}, \, g_4 = -\frac{19}{720}, \text{ and } g_5 = \frac{3}{160}.$$

In the Fig 1, we describe the image behavior of Gregory coefficients $\varphi(t)$ under the unit disk.

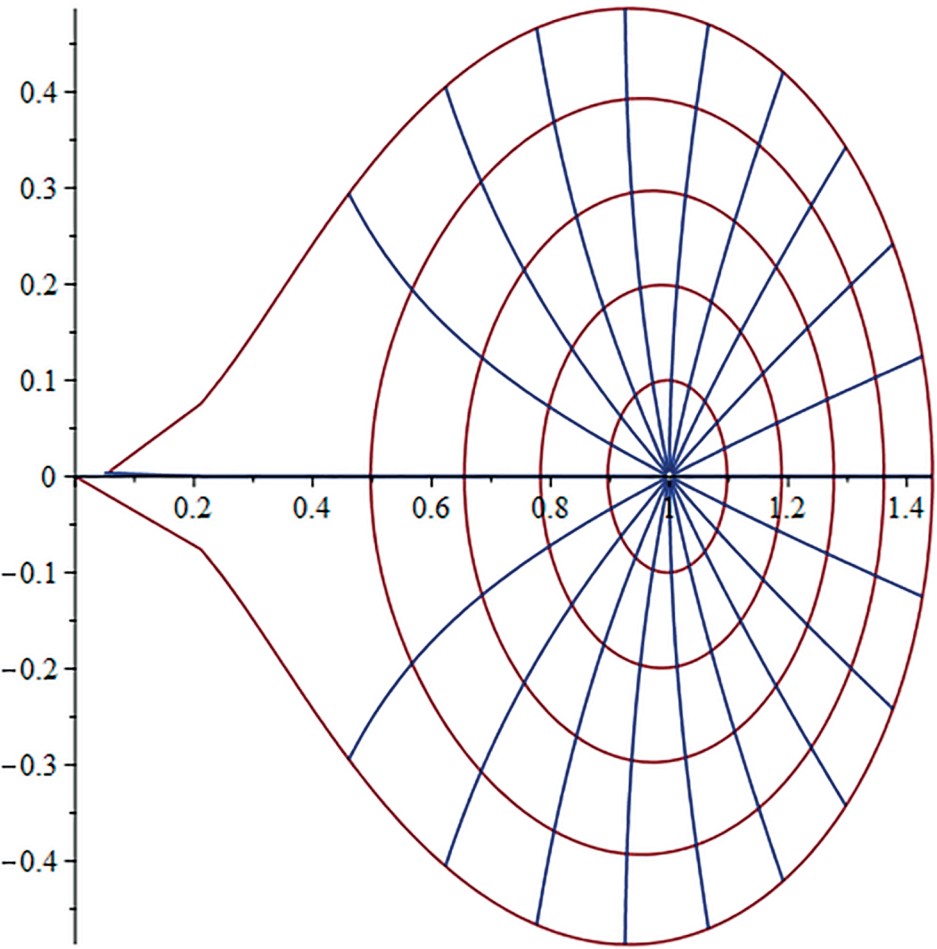

**Fig 1. The image of $\varphi(t)$ on the unit disk.**

Recently, Bulut [49] research focused on applying Faber polynomial methods to analytic bi-univalent functions associated with Gregory coefficients, introducing the following class definition:

**Definition 1.** [49]. For $1 \leq \xi, 0 \leq \upsilon$. A function $h \in \Sigma$ defined in (1) is in the class $\mathcal{G}_{\Sigma}^{\xi, \upsilon}(\varphi)$ if

$$(1 - \xi)\left(\frac{h(t)}{t}\right)^{\upsilon} + \xi h^{'}(t)\left(\frac{h(t)}{t}\right)^{\upsilon-1} \quad \prec \quad \varphi(t),$$

$$(1 - \xi)\left(\frac{g(w)}{w}\right)^{\upsilon} + \xi g^{'}(w)\left(\frac{g(w)}{w}\right)^{\upsilon-1} \quad \prec \quad \varphi(w),$$

where $\Sigma$ denote the class of bi-univalent functions, $\varphi(t)$ is defined by (7) and $g(w) = h^{-1}(t)$.

Murugusundaramoorthy et al. [50] investigated three classes of bi-univalent functions related to Gregory coefficients, defining them as follows:

**Definition 2.** [50]. A function $h \in \Sigma$ defined in (1) is in the class $\mathfrak{H}\mathfrak{G}_\Sigma$ if

$$
\begin{aligned}
h^{'}(t) &\prec \varphi(t), \\
g^{'}(w) &\prec \varphi(w),
\end{aligned}
$$

where $\Sigma$ denote the class of bi-univalent functions and $g(w) = h^{-1}(t)$.

**Definition 3.** [50]. For $0 \leq \mu \leq 1$. A function $h \in \Sigma$ defined in ( 1) is in the class $\mathfrak{G}\mathfrak{M}_\Sigma(\mu)$ if

$$
\begin{aligned}
(1-\mu)\frac{th^{'}(t)}{h(t)} + \mu\left(1 + \frac{th^{''}(t)}{h^{'}(t)}\right) &\prec \varphi(t), \\
(1-\mu)\frac{wg^{'}(w)}{g(w)} + \mu\left(1 + \frac{wg^{''}(t)}{g^{'}(t)}\right) &\prec \varphi(w),
\end{aligned}
$$

where $\Sigma$ denote the class of bi-univalent functions and $g(w) = h^{-1}(t)$.

**Definition 4.** [50]. For $\gamma \in (-\pi, \pi]$. A function $h \in \Sigma$ defined in (1) is in the class $\mathfrak{G}_\Sigma(\gamma)$ if

$$
\begin{aligned}
\frac{th^{'}(t)}{h(t)} + \frac{1+e^{i\gamma}}{2}\frac{t^2h^{''}(t)}{h(t)} &\prec \varphi(t), \\
\frac{wg^{'}(w)}{g(w)} + \frac{1+e^{i\gamma}}{2}\frac{w^2g^{''}(t)}{g(t)} &\prec \varphi(w),
\end{aligned}
$$

where $\Sigma$ denote the class of bi-univalent functions and $g(w) = h^{-1}(t)$.

Firstly, they demonstrated that these classes are non-empty. Furthermore, for functions within each of these three bi-univalent function classes, Murugusundaramoorthy et al. [50] investigated the initial estimates $|a_2|$ and $|a_3|$ of the Taylor–Maclaurin coefficients and Fekete–Szego $\left|a_3 - \eta a_2^2\right|$ functional problems. Building on the aforementioned research, we now define a new class of symmetric starlike functions, denoted by $\mathcal{S}_{S,G}^*$, which are related to the symmetric points associated with Gregory coefficients. This class is characterized by the following properties:

**Definition 5.** Suppose that the function $h \in \mathcal{S}_{S,G}^*$, as defined by the Eq (1), if

$$
\frac{th^{'}(t)}{h(t) - h(-t)} \prec \varphi(t), \tag{8}
$$

where $\varphi(t)$ is given in (7).

## 2. Set of Lemmas

**Lemma 1.** ([51],[52]). Suppose the function $p$ belongs to the class $\mathcal{P}$, and is defined by Eq (2). Then

$$
|c_n| \leq 2, \tag{9}
$$

$$
|c_{n+k} - \mu c_n c_k| \leq 2, \quad \text{if } 0 \leq \mu \leq 1, \tag{10}
$$

$$|c_m c_n - c_k c_l| \leq 4, \quad \text{if } m + n = k + l, \tag{11}$$

$$\left| c_{n+2k} - \mu c_n c_k^2 \right| \leq 2 \left( 1 + 2\mu \right), \quad \text{for } \mu \in \mathbb{R}, \tag{12}$$

$$\left| c_2 - \frac{|c_1^2|}{2} \right| \leq 2 - \frac{|c_1^2|}{2}. \tag{13}$$

**Lemma 2.** [53]. Suppose the function $p$ belongs to the class $\mathcal{P}$, and is defined by Eq (2). Then

$$4c_3 = c_1^3 + 2(4 - c_1^2)c_1 x - c_1(4 - c_1^2)x^2 + 2(4 - c_1^2)1 - |x|2y \tag{14}$$

and

$$2c_2 = c_1^2 + x(4 - c_1^2) \tag{15}$$

for some $x, y \in C$ with $|x| \leq 1$ and $|y| \leq 1$.

**Lemma 3.** Suppose the function $p \in \mathcal{P}$, and is defined by Eq (2), then

$$|c_n| \leq 2, \quad \text{if } n \geq 1, \tag{16}$$

and if $F \in [0, 1]$ and $F(2F - 1) \leq E \leq F$, then

$$\left| c_3 - 2Fc_1 c_2 + Ec_1^3 \right| \leq 2. \tag{17}$$

For complex numbers $\lambda$ we have

$$\left| c_2 - \lambda c_1^2 \right| \leq 2 \max \left\{ 1, |2\lambda - 1| \right\}. \tag{18}$$

See [52–56] for the inequality (16), (17) and ( 18).

**Lemma 4.** [57]. Suppose the function $p$ belongs to the class $\mathcal{P}$, and is defined by Eq (2), $0 < T_2 < 1$, $0 < Q_1 < 1$ and

$$8T_2(1 - T_2) \left[ (Q_1 Q_2 - 2T_1)^2 + (Q_1(T_2 + Q_1) - Q_2)^2 \right] + (Q_2 - 2T_2 Q_1)^2 Q_1(1 - Q_1)$$
$$\leq \quad 4Q_1^2 T_2(1 - Q_1)^2(1 - T_2). \tag{19}$$

Then

$$\left| T_1 c_1^4 + T_2 c_2^2 + 2Q_1 c_1 c_3 - \frac{3}{2} Q_2 c_1^2 c_2 - c_4 \right| \leq 2. \tag{20}$$

**Lemma 5.** [58] Let $U = \{t : |t| \leq 1\}$. Also, for $R, L, M \in \mathbb{R}$, let $Y(R, L, M) = \max \left\{ \left| R + Lx + Mx^2 \right| \right.$ $+1 - |x|^2, \ x \in U \right\}$. If $RM \geq 0$, then

$$Y(R, L, M) = \left\{ \begin{array}{ll} |R| + |L| + |M| & L \geq 2(1 - |M|), \\ 1 + |R| + \frac{L^2}{4(1 - |M|)} & |L| < 2(1 - |M|). \end{array} \right\}$$

Furthermore, if $RM<0$, then

$$Y(R,L,M) = \left\{ \begin{array}{ll} 1 - |R| + \frac{L^2}{4(1-|M|)} & \left(-4RM(M^{-2}-1) \le L^2; |L| < 2(1-|M|)|\right) \\ 1 + |R| + \frac{L^2}{4(1+|M|)} & \left(|L^2 < \min\left\{4(1+|M|)^2; -4RM(M^{-2}-1)\right\}\right) \\ R(R,L,M) & otherwise \end{array} \right\},$$

where

$$Y(R,L,M) = \left\{ \begin{array}{ll} |R| + |L| - |M| & (|M|(|L|+4|R|) \le |RL|)) \\ -|R| + |L| + |M| & (|RL| \le |M|(|L|-4|R|))) \\ (|R|+|M|)\sqrt{1-\frac{L^2}{4RM}}, & otherwise \end{array} \right\}$$

## 3. Main results

In the following result, we establish initial bounds for the function $h \in \mathcal{S}_{S,G}^*$.

**Theorem 1.** Assume the function $h$, defined by (1), is in the class $\mathcal{S}_{S,G}^*$, then

$$|a_2| \le \frac{1}{4}, \; |a_3| \le \frac{1}{6}, \; |a_4| \le \frac{1}{4}, \; |a_5| \le \frac{1}{8}.$$

The estimates provided are sharp and achieved by the functions given in (29)–(32), respectively.

**Proof:** Given that $h$ is a member of $\mathcal{S}_{S,G}^*$, and using the definition of the Schwarz function, we obtain

$$\frac{th'(t)}{h(t) - h(-t)} = \Psi(u(t)) = \frac{u(t)}{\ln(1 + u(t))}. \tag{21}$$

The function $p$ is defined as

$$p(t) = 1 + c_1 t + c_2 t^2 + c_3 t^3 + ...,$$

then $p \in \mathcal{P}$. This implies that

$$u(t) = \frac{c_1 t + c_2 t^2 + c_3 t^3 + ...}{2 + c_1 t + c_2 t^2 + c_3 t^3 + ...}. \tag{22}$$

It is evident that $p$ is analytic in the region $U$, satisfying $p(0) = 1$, and $Rep(t) > 0$. By using (22) and $\frac{w(t)}{\ln(1+u(t))}$, we get

$$\begin{aligned} \Psi(u(t)) &= 1 + \frac{1}{4}c_1 t + \frac{1}{48}\left(12c_2 - 7c_1^2\right)t^2 + \frac{1}{192}\left(17c_1^3 - 56c_1 c_2 + 48c_3\right)t^3 \\ &\quad + \frac{1}{11520}\left(-649c_1^4 + 3060c_1^2 c_2 - 3360c_1 c_3 - 1680c_2^2 + 2880c^4\right)t^4 \\ &\quad + \frac{1}{46080}(1739c_1^5 - 10384c_1^3 c_2 + 12240c_1^2 c_3 + 12240c_1 c_2^2 \\ &\quad - 13440c_2 c_3 + 13440c_1 c_4 + 11520c_5)t^5... \end{aligned}$$

and

$$\frac{th^{'}(t)}{h(t) - h(-t)} = 1 + 2a_2t + 3a_3t^2 + \left(4a_4 - 2a_2a_3\right)t^3 + \left(4a_5 - 2a_3^2\right)t^4 + \dots .\tag{23}$$

It follows by (21), (22) and (23) that

$$a_2 = \frac{c_1}{8},\tag{24}$$

$$a_3 = \frac{1}{12}\left(c_2 - \frac{7}{12}c_1^2\right),\tag{25}$$

$$a_4 = \frac{1}{8}\left(c_3 - \frac{39}{36}c_1c_2 + \frac{11}{36}c_1^3\right),\tag{26}$$

$$a_5 = \frac{1}{16}\left(-\frac{5351}{25920}c_1^4 - \frac{19}{36}c_2^2 - \frac{7}{6}c_1c_3 + \frac{431}{444}c_1^2c_2 + c^4\right).\tag{27}$$

Using the inequality (9) of Lemma 1, on $a_2$, we get

$$|a_2| \leq \frac{1}{4}.$$

Rearrange (25), we have

$$|a_3| = \frac{1}{12}\left|c_2 - \frac{7}{12}c_1^2\right|.\tag{28}$$

Using the inequality (10) of Lemma 1, on (28), we have

$$|a_3| \leq \frac{1}{6}.$$

Rearranging (26) and (27) it gives

$$\begin{aligned}|a_4| &= \frac{1}{8}\left|c_3 - \frac{39}{36}c_1c_2 + \frac{11}{36}c_1^3\right| \\ &= \frac{1}{8}\left|c_3 - 2\left(\frac{39}{72}\right)c_1c_2 + \frac{11}{36}c_1^3\right| \\ &= \frac{1}{8}\left|c_3 - 2Fc_1c_2 + Ec_1^3\right|\end{aligned}$$

where

$$F = \frac{39}{72} \text{ and } E = \frac{11}{36}.$$

It gives us $0 < F < 1$, $E \leq F$ and $F(2F - 1) \leq E \leq F$. Therefore by using the Lemma 3, we have

$$|a_4| \leq \frac{1}{4}.$$

Again rearrange (27) we have

$$
\begin{aligned}
|a_5| &= \frac{1}{16}\left|\frac{5351}{25920}c_1^4 + \frac{19}{36}c_2^2 + \frac{7}{6}c_1c_3 - \frac{431}{444}c_1^2c_2 - c_4\right| \\
&= \frac{1}{16}\left|\frac{5351}{25920}c_1^4 + \frac{19}{36}c_2^2 + 2\left(\frac{7}{12}\right)c_1c_3 - \frac{3}{2}\left(\frac{431}{666}\right)c_1^2c_2 - c_4\right| \\
&= \frac{1}{16}\left|T_1c_1^4 + T_2c_2^2 + 2Q_1c_1c_3 - \frac{3}{2}Q_2c_1^2c_2 - c_4\right|,
\end{aligned}
$$

where

$$
T_1 = \frac{5351}{25920},\ T_2 = \frac{19}{36},\ Q_1 = \frac{7}{12}, Q_2 = \frac{431}{666}.
$$

Now we have

$$
8T_2(1-T_2)\left[(Q_1Q_2-2T_1)^2 + (Q_1(T_2+Q_1)-Q_2)^2\right] + (Q_2-2T_2Q_1)^2Q_1(1-Q_1) = 0.02743900
$$

and

$$
4Q_1^2T_2(1-Q_1)^2(1-T_2) = 0.05890.
$$

Hence (19) of the Lemma 4 is satisfy, therefore using the inequality (20) of Lemma 4, we have

$$
|a_5| \le \frac{1}{8}.
$$

The bounds, $|a_2|$, $|a_3|$, $|a_4|$ and $|a_5|$ are sharp for the following extremal functions:

$$
\frac{th'(t)}{h(t)-h(-t)} = \frac{t}{\ln(1+t)} = 1 + \frac{1}{2}t + ..., \tag{29}
$$

$$
\frac{th'(t)}{h(t)-h(-t)} = \frac{t^2}{\ln(1+t^2)} = 1 + \frac{1}{2}t^2 + ..., \tag{30}
$$

$$
\frac{th'(t)}{h(t)-h(-t)} = \frac{t^3}{\ln(1+t^3)} = 1 + \frac{1}{2}t^3 + ..., \tag{31}
$$

$$
\frac{th'(t)}{h(t)-h(-t)} = \frac{t^4}{\ln(1+t^4)} = 1 + \frac{1}{2}t^4 + .... \tag{32}
$$

Therefore, the proof is now complete.

**Theorem 2.** Suppose that $h \in \mathcal{S}_{S,G}^*$. Then, the following sharp estimates hold:

$$
|a_3 - \mu a_2^2| \le \frac{1}{6}\max\left\{1, \left|\frac{4+9\mu}{24}\right|\right\},\ \mu \in \mathbb{C}.
$$

Theorem 2 is sharp for the function defined in (30).

**Proof:** Using (24) and (25), we obtain

$$
|a_3 - \mu a_2^2| = \frac{1}{12}\left|c_2 - \frac{28+9\mu}{48}c_1^2\right|.
$$

Using the inequality (18) of Lemma 3, we have

$$|a_3 - \mu a_2^2| \le \frac{1}{6} \max \left\{ 1, \left| \frac{4 + 9\mu}{24} \right| \right\} \quad \mu \in \mathbb{C}.$$

Therefore, the proof is now complete.

**Corollary 1.** Suppose $h \in \mathcal{S}_{S,G}^*$. Then

$$|a_3 - a_2^2| \le \frac{1}{6}.$$

The estimate is sharp for the function provided in Eq (30).

**Theorem 3.** Suppose $h \in \mathcal{S}_{S,G}^*$. Then

$$|a_2 a_3 - a_4| \le \frac{1}{4}. \tag{33}$$

Theorem 3 is sharp for the function provided in Eq (29).

**Proof:** Using (24), (25) and (26), we get

$$
\begin{aligned}
|a_2 a_3 - a_4| &= \frac{1}{8} \left| c_3 - \frac{21}{144} c_1 c_2 + \frac{17}{384} c_1^3 \right|, \\
&= \frac{1}{8} \left| c_3 - 2 \left( \frac{21}{288} \right) c_1 c_2 + \frac{17}{384} c_1^3 \right|, \\
&= \frac{1}{8} \left| c_3 - 2 F c_1 c_2 + E c_1^3 \right|,
\end{aligned}
$$

where

$$F = \frac{21}{288} \text{ and } E = \frac{17}{384}.$$

It gives us $0 < F < 1$, $E < F$ and $F(2F - 1) < E < F$. Therefore by using the inequality (17) of Lemma 3, we have

$$|a_2 a_3 - a_4| \le \frac{1}{4}.$$

Therefore, the proof is now complete.

**Theorem 4.** Suppose that $h \in \mathcal{S}_{S,G}^*$. Then

$$|a_2 a_4 - a_3^2| \le \frac{1}{36}.$$

The estimate is sharp for the function provided in Eq (30).

**Proof:** From (24), (25) and (26), we have

$$|a_2 a_4 - a_3^2| = \frac{1}{20736} \left| 50 c_1^4 - 183 c_1^2 c_2 + 324 c_1 c_3 - 144 c_2^2 \right|.$$

Applying the Lemma 2 and let $s = c_1 \in [0, 2]$, we can write

$$
\begin{aligned}
\left|a_2 a_4 - a_3^2\right| &= \frac{1}{41472} \left\{ -43s^4 - 3s^2 \left(4 - s^2\right) x - \left(4 - s^2\right) \right. \\
&\qquad \left. \left\{162s^2 + 72\left(4 - s^2\right)\right\} x^2 + 324s\left(4 - s^2\right)\left(1 - |x|^2\right) y \right\} \\
&= T.
\end{aligned}
$$

Since $|x| \leq 1$ and if $s = 0$, then $T = \frac{-1}{36} x^2$. Therefore

$$
|T| \leq \frac{1}{36}.
$$

If $s = 2$, then

$$
|T| \leq \frac{43}{162}. \tag{34}
$$

Suppose that $s \in (0, 2)$. Then, we have

$$
\begin{aligned}
|T| &= \frac{s\left(4 - s^2\right)}{128} \left| \frac{-43s^3}{324\left(\left(4 - s^2\right)\right)} - \frac{3s}{324} x - \left(\frac{288 + 162s - 72s^2}{324s}\right) x^2 + \left(1 - |x|^2\right) y \right| \\
&\leq \frac{s\left(4 - s^2\right)}{128} \left| \frac{-43s^3}{324\left(4 - s^2\right)} - \frac{3s}{324} x - \left(\frac{288 + 162s - 72s^2}{324s}\right) x^2 \right| + \left(1 - |x|^2\right) \\
&= \frac{s\left(4 - s^2\right)}{128} \left[ \left|R + Lx + Mx^2\right| + 1 - |x|^2 \right],
\end{aligned}
$$

where

$$
R = \frac{-43s^3}{324\left(\left(4 - s^2\right)\right)}, \quad L = -\frac{3s}{324}, \quad M = -\left(\frac{288 + 162s - 72s^2}{324s}\right).
$$

Consequently, $RM > 0$. Furthermore, it is readily apparent that

$$
|L| - 2(1 - |M|) = \frac{11s^2 - 36s + 32}{18s} > 0 \ \text{ for } s \in (0, 2) .
$$

Thus, we get

$$
\begin{aligned}
|T| &\leq \frac{s\left(4 - s^2\right)\left(|R| + |L| + |M|\right)}{128} \\
&= \frac{\left(4 - s^2\right)}{41472} \left(43s^4 + 108s^2 + 288\right), \\
&= H_0(t).
\end{aligned}
$$

Let $l = s^2$, and $l \in (0, 4)$. Then

$$
H_0(l) = \frac{-43l^3 + 64l^2 + 144l}{41472}.
$$

In that case, we have

$$|T| \le H_0(t) \le \frac{-43l^3 + 64l^2 + 144l}{41472} \le \frac{1}{36}. \tag{35}$$

Therefore, the proof is now complete.

**Theorem 5.** Assume that $h \in \mathcal{S}_{S,G}^*$. Then

$$|H_3(1)| \le \frac{19}{216}.$$

**Proof:** Since by (6), we have

$$H_{3,1}(h) = a_5(a_3 - a_2^2) + a_3(a_2a_4 - a_2^3) - a_4(a_4 - a_2a_3).$$

By using Theorem 1, Theorem 1, Theorem 3 and Theorem 4, we have

$$|H_3(1)| \le |a_4||a_4 - a_2a_3| + |a_3||a_2a_4 - a_2^3| + |a_5||a_3 - a_2^2| \le \frac{19}{216}.$$

Therefore, the proof is now complete.

**Theorem 6.** Assume the function $h$, defined by (1), belongs to $\mathcal{S}_{S,G}^*$ and has the power series representation $h^{-1}(w) = w + A_2w^2 + A_3w^3 + ...$, then

$$|A_2| \le \frac{1}{4}, \ |A_3| \le \frac{1}{6}, \ |A_4| \le \frac{1}{4}. \tag{36}$$

The estimates provided are sharp and achieved by the functions given in (29)-(31), respectively.

**Proof:** If $h \in \mathcal{S}_{S,G}^*$ and $h^{-1}(w) = w + A_2w^2 + A_3w^3 + ...$. Substituting Eq (24) into equation (5), we get

$$A_2 = -\frac{c_1}{8},$$

Using the inequality (9) of Lemma 1, on $A_2$, we have

$$|A_2| \le \frac{1}{4}.$$

Now for $A_3$, substituting Eqs (24), (25) in (5), we have

$$\begin{aligned} A_3 &= 2a_2^2 - a_3 \\ &= \frac{-1}{12}\left(c_2 - \frac{23}{24}c_1^2\right). \end{aligned}$$

Thus

$$|A_3| = \frac{1}{12}\left|c_2 - \frac{23}{24}c_1^2\right|.$$

By the inequality (10) of Lemma 1, we have

$$|A_3| \le \frac{1}{6}.$$

Now for $A_4$, so from (5), we have

$$
\begin{aligned}
|A_4| &= \frac{1}{8}\left|c_3 - \frac{3}{2}c_1c_2 + \frac{361}{576}c_1^3\right| \\
&= \frac{1}{8}\left|c_3 - 2\left(\frac{3}{4}\right)c_1c_2 + \frac{361}{576}c_1^3\right| \\
&= \frac{1}{8}\left|c_3 - 2Fc_1c_2 + Ec_1^3\right|,
\end{aligned}
$$

where

$$
F = \frac{3}{4} \text{ and } E = \frac{361}{576}.
$$

It gives us $0 < F < 1$, $E < F$ and $F(2F - 1) < E < F$. Therefore by using the inequality (17) of Lemma 3, we have

$$
|A_4| \leq \frac{1}{4}.
$$

Therefore, the proof is now complete.

**Logarithmic function**

The logarithmic coefficients $L_n$ of a function $h(t) = t + \sum_{n=2}^{\infty} a_n t^n$, which belongs to the class $\mathcal{S}$, are defined by the following formula:

$$
G_h(t) = \log\left(\frac{h(t)}{t}\right) = 2\sum_{n=2}^{\infty} L_n t^n \quad \text{for } t \in U. \tag{37}
$$

The logarithmic coefficients have far-reaching implications in the study of univalent functions, and their impact is evident in numerous estimates. De Branges [59] seminal work in 1985 demonstrated that

$$
\sum_{k=1}^{n} k(n-k+1)|L|^2 \leq \sum_{k=1}^{n} \frac{n-k+1}{k} \quad n \geq 1 \tag{38}
$$

and the equality holds if and only if $h$ has the specific form $t/(1 - e^{i\theta})^2$, where $\theta$ is a real number.

This inequality is a cornerstone of univalent function theory, encompassing the celebrated Bieberbach-Robertson-Milin conjectures on Taylor coefficients. Andreev and Duren [60] notably employed logarithmic coefficients to establish Brennan's conjecture for conformal mappings. The study of logarithmic coefficients has since flourished, with significant contributions from, Alimohammadi et al. [61], Deng [62], Roth [63], Ye [64], and Girela [65]. Their work has substantially expanded our knowledge of logarithmic coefficients in various subclasses of holomorphic univalent functions. According to the definition, the logarithmic coefficients for a function $h$ in $\mathcal{S}$ are easily calculated as:

$$
L_1 = \frac{a_2}{2}, \tag{39}
$$

$$
L_2 = \frac{1}{2}\left(a_3 - \frac{1}{2}a_2^2\right), \tag{40}
$$

$$L_3 = \frac{1}{2}\left(a_4 - a_2 a_3 - \frac{1}{2}a_2^3\right), \tag{41}$$

$$L_4 = \frac{1}{2}\left(a_5 - a_2 a_4 + a_2 a_3^2 - \frac{1}{2}a_3^2 - \frac{1}{4}a_2^4\right). \tag{42}$$

**Theorem 7.** Assume that $h \in \mathcal{S}_{S,G}^*$. Then

$$|L_1| \le \frac{1}{8}, \ |L_2| \le \frac{1}{12}, |L_3| \le \frac{1}{8} \text{ and } |L_4| \le \frac{1}{16}. \tag{43}$$

**Proof:** Let $h \in \mathcal{S}_{S,G}^*$. Then, using the Eqs (24), (25), (26) and (27), in (39), (40), (41) and (42), we get

$$L_1 = \frac{c_1}{16}, \tag{44}$$

$$L_2 = \frac{1}{24}\left(c_2 - \frac{65}{96}c_1^2\right), \tag{45}$$

$$L_3 = \frac{1}{16}\left(c_3 - \frac{42}{36}c_1 c_2 + \frac{207}{576}c_1^3\right) \tag{46}$$

and

$$L_4 = \frac{-1}{32}\left(\frac{34407}{116642}c_1^4 + \frac{17}{12}c_1 c_3 + \frac{19}{36}c_2^2 - \frac{3363}{2664}c_1^2 c_2 - c_4\right). \tag{47}$$

For $L_1$, using the inequality (9) of Lemma 1 , on (44), we obtain

$$|L_1| \le \frac{1}{8}.$$

For $L_2$, rearrange (45), thus

$$|L_2| = \frac{1}{24}\left|c_2 - \frac{65}{96}c_1^2\right|.$$

Using the inequality (10) of Lemma 1, we have

$$|L_2| \le \frac{1}{12}.$$

For $L_3$, rearrange (46) as:

$$\begin{aligned}
|L_3| &= \frac{1}{16}\left|c_3 - \frac{42}{36}c_1 c_2 + \frac{207}{576}c_1^3\right| \\
&= \frac{1}{16}\left|c_3 - 2\left(\frac{7}{12}\right)c_1 c_2 + \frac{207}{576}c_1^3\right| \\
&= \frac{1}{16}\left|c_3 - 2F c_1 c_2 + E c_1^3\right|,
\end{aligned}$$

where

$$F = \frac{7}{12} \text{ and } E = \frac{207}{576}.$$

It gives us $0<F<1$, $E<F$ and $F(2F-1)<E<F$. Therefore by using the inequality (17) of Lemma 3, we have

$$|L_3| \leq \frac{1}{8}.$$

For $L_4$, rearrange (47) as:

$$
\begin{aligned}
L_4 &= \frac{1}{32}\left|\frac{649}{2880}c_1^4 + \frac{7}{12}c_2^2 + \frac{7}{6}c_1c_3 - \frac{17}{16}c_1^2c_2 - c_4\right| \\
&= \frac{1}{32}\left|\frac{649}{2880}c_1^4 + \frac{7}{12}c_2^2 + \frac{7}{6}c_1c_3 - \frac{17}{16}c_1^2c_2 - c_4\right| \\
&= \frac{1}{32}\left|\rho c_1^4 + \varsigma c_2^2 + 2\delta c_1c_3 - \frac{3}{2}\psi c_1^2c_2 - c_4\right|,
\end{aligned}
$$

where

$$\rho = \frac{649}{2880}, \varsigma = \frac{7}{12}, \delta = \frac{7}{12} \text{ and } \psi = \frac{17}{24}.$$

It follows that

$$8\varsigma(1-\varsigma)\left[(\delta\psi - 2\rho)^2 + (\delta(\varsigma+\delta) - \psi)^2\right] + \delta(1-\delta)(\psi - 2\varsigma\delta)^2 = 0.0746042647835016$$

and

$$4\varsigma\delta^2(1-\delta)^2(1-\varsigma) = 0.0848068844307270233.$$

Hence (19) of the Lemma 4 is satisfy, therefore using the inequality (20) of Lemma 4, we have

$$|L_4| \leq \frac{1}{16}.$$

Therefore, the proof is now complete.

## 4. Conclusion

This research has established a new class, $\mathcal{S}_{S,G}^*$, of symmetric starlike functions, which are connected to the generating function of Gregory coefficients through a subordination relationship. Our investigation has led to the discovery of various coefficient inequalities, including sharp coefficient bounds, Fekete-Szego problems and an upper bound for the third-order Hankel determinant, and inverse inequalities. Finally, we have also derived sharp estimates for the logarithmic and inverse coefficients of functions in the class $\mathcal{S}_{S,G}^*$ of symmetric starlike functions. Future research directions include further exploration of the class $\mathcal{S}_{S,G}^*$ to determine Toeplitz and higher-order Hankel determinants, Extreme point theorem, Partial sums results, Necessary and sufficient conditions, Convex combination, Closure theorem, Growth and distortion bounds, Radii of close-to-starlikeness and starlikeness.

## Author contributions

**Conceptualization:** Mohammad Faisal Khan.

**Data curation:** Mohammad Faisal Khan, Muqrin A. Almuqrin.

**Formal analysis:** Muqrin A. Almuqrin.

**Funding acquisition:** Mohammed Abaoud.

**Investigation:** Muqrin A. Almuqrin.

**Methodology:** Mohammad Faisal Khan.

**Project administration:** Mohammed Abaoud.

**Resources:** Mohammed Abaoud.

**Software:** Mohammed Abaoud, Muqrin A. Almuqrin.

**Supervision:** Mohammad Faisal Khan.

**Validation:** Naeem Ahmad.

**Writing – original draft:** Naeem Ahmad.

**Writing – review & editing:** Naeem Ahmad.

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
