## [Decision Letter · Decision Letter 0]

8 Oct 2024

PONE-D-24-33833New class of symmetric starlike functions subordinate to the Generating function of Gregory coefficientsPLOS ONE

Dear Dr. Khan,

Thank you for submitting your manuscript to PLOS ONE. After careful consideration, we feel that it has merit but does not fully meet PLOS ONE’s publication criteria as it currently stands. Therefore, we invite you to submit a revised version of the manuscript that addresses the points raised during the review process.

We look forward to receiving your revised manuscript.

Kind regards,

Mohamed Kamel Riahi

Academic Editor

PLOS ONE

“The authors extend the appreciation to the Deputyship for research and Innovation, Ministry of Education Saudi Arabia for funding this research through the project number IFP-IMSIU-2023137. The authors also appreciate the Deanship of Scientific Research at Imam Mohammad Ibn Saud Islamic University (IMSIU) for supporting and supervising this project.”

4. Please remove your figure from within your manuscript file, leaving only the individual TIFF/EPS image files, uploaded separately. These will be automatically included in the reviewers’ PDF.

Additional Editor Comments:

After careful consideration of the independent reviewers’ reports, we would like to request a minor revision before moving forward with the publication process. The reviewers have recognized the value of your work but have highlighted a few areas that need clarification or improvement.

Reviewers' comments:

Reviewer's Responses to Questions

**Comments to the Author**

1. Is the manuscript technically sound, and do the data support the conclusions?

Reviewer #1: Yes

Reviewer #2: Yes

2. Has the statistical analysis been performed appropriately and rigorously? 

Reviewer #1: Yes

Reviewer #2: N/A

3. Have the authors made all data underlying the findings in their manuscript fully available?

Reviewer #1: Yes

Reviewer #2: Yes

4. Is the manuscript presented in an intelligible fashion and written in standard English?

Reviewer #1: Yes

Reviewer #2: Yes

5. Review Comments to the Author

Reviewer #1: Title: New class of symmetric starlike functions subordinate to the generating functions of Gregory coefficients

Report: The paper " New class of symmetric starlike functions subordinate to the generating functions of Gregory coefficients " by Khan et al. presents an innovative exploration into the realm of geometric function theory through the application of symmetric functions. The authors introduced a class of symmetric starlike functions subordinate to the generating functions of Gregory coefficients. This study uses a specific mathematical technique (subordination) to define a new class of analytic and symmetric starlike functions. The paper demonstrates rigorous mathematical derivations and proofs, ensuring the correctness and reliability of the results. The authors meticulously derived and explored new findings, such as sharp bounds for problem concerning the coefficients of the family of symmetric starlike functions connected to the generating function of Gregory coefficients by utilizing the notion of functions with positive real component. These problems include first five sharp coefficients bounds and Fekete-Szego problem along with the Hankel determinant of order three. Additionally, they explored the optimal bounds (sharp bounds) for two important functions, the logarithmic function and the inverse function within the same class of symmetric starlike functions.

The study not only introduces new classes but also explores sharp functions for all results and figure 1 show the image of unit disk under the functions of Gregory coefficients. The use of sharp functions and graphical studies adds substantial value to this article. The manuscript is well-organized, with clear definitions, theorems, and proofs. The mathematical formulations and derivations presented are precise and well-founded. The introduction comprehensively covers all the necessary concepts related to this work. Abstract and conclusions covers all reseach work and written clearly. Based on a thorough review, the mathematical procedures appear correct. Overall, the paper presents a good analytical idea and just needs a check for typos and grammar.

Here are some suggested changes:

1. On page 1, line 5: remove the extra “can” from the following sentence” A can can be written as:”

2. On page 4, line 17: Replace “precise bound” with “sharp bounds”. We cannot write precise bounds.

3. On page 4: write “ and “ betwee the equations on line 22 and 23.

4. On page 5: Remove the period in Definition 1.

5. On page 6: In the last line of Lemma 2, it should be like this|x|≤1,|y|≤1.

6. On page 6: In the second line of Lemma 4, replace 0<a<1 0="" with=""> 7. On page 7, line 8: Remove “period” and put the “ comma” after (3.9)-(3.12)

8. On page 11, line 25: “By the inequality (??) of Lemma 3, we have”. Equation number is missing.

9. On page 13, line 6: replace "v" with " using".

10. On page 14, line 2: remove the period

11. The conclusion section lacks clarity on future directions. Please revise and expand the conclusions to include specific future research directions

I recommend making these corrections and thoroughly reviewing the entire manuscript for typo error.

Decision: In summary, the paper by Khan et al. makes a significant contribution to the field of symmetric starlike functions and Geometric Functions Theory. The research work, "New class of symmetric starlike functions subordinate to the generating functions of Gregory coefficients," represents a noteworthy achievement. The paper's mathematical rigor and clear presentation make it a valuable resource for researchers and practitioners in the field.

The article can be published after implementing these minor changes.</a<1>

Reviewer #2: \item The authors advised to use the variable as z, u, or t not as $\gamma$ from 91.1) onwards.

\item There is no novelty why author consider this study and as usual routine procedure they obtained the results.

\item check for typos completely for upper case, lower case, and pulley stop , etc.

\item The results are correctly formulated and could be understand easily;

\item The proofs are correct and clearly presented, and there are no any mistakes in the proofs.

The authors are advised to add few recent works on nephroid domain and Lemiscate domaina, also the references given below

\item A class of -bi-pseudo-starlike functions with respect to symmetric points associated with Telephone numbers. Afr. Mat. 35, 17 (2024). https://doi.org/10.1007/s13370-023-01159-0.

\\

The authors purposely avoided the following , they must add these articles which are more related to their study.

\item Initial Coefficient Bounds for Bi-Univalent Functions Related to Gregory Coefficients Mathematics 2023, 11(13), 2857; https://doi.org/10.3390/math11132857

\item Korean J. Math.32(2024), No. 2, pp. 285–295https://dx.doi.org/10.11568/kjm.2024.32.2.285FABER POLYNOMIAL COEFFICIENT ESTIMATES FORANALYTIC BI-UNIVALENT FUNCTIONS ASSOCIATED WITH GREGORY COEFFICIENTS

6. PLOS authors have the option to publish the peer review history of their article (what does this mean?). If published, this will include your full peer review and any attached files.

Reviewer #1: **Yes: **Shahid Khan

Reviewer #2: **Yes: **G.Murugusundaramoorthy

---

## [Author Response · Author response to Decision Letter 1]

17 Oct 2024

Reviewer 1

I would like to extend my sincerest gratitude to the editor and reviewers for their meticulous evaluation, insightful suggestions, and constructive comments, which significantly enhanced the quality and clarity of this article.

Reply to the reviewer 1:

The authors advised to use the variable as z, u, or t not as γ from 91.1 onwards.

Reply: In accordance with the reviewers' feedback, we have replaced the variable γ with t throughout the revised manuscript.

There is no novelty why author consider this study and as usual routine procedure they obtained.

Reply: Respected Professor: Regarding point 2, I would like to elaborate on a crucial aspect. In the revised version, I have highlighted this point. Recently, researchers explored classes of starlike, convex, and bounded turning functions using Generating functions of Gregory coefficients. However, our article introduces a novel class of starlike functions of symmetric points, also utilizing Generating functions of Gregory coefficients. Specifically, we investigate sharp inequalities, Hankel Determinant, logarithmic, and inverse problems.

Check for typos completely for upper case, lower case, and pulley stop , etc.

Reply: We thoroughly reviewed the text to remove typos and other errors.

The proofs are correct and clearly presented, and there are no any mistakes in the proofs. The authors are advised to add few recent works on nephroid domain and Lemiscate domaina, also the references given below.

Reply: Respected Professor, we incorporated new recent works into the revised version, as per your suggestions, and highlighted them in the PDF version for clarity.

8. A class of -bi-pseudo-starlike functions with respect to symmetric points associated with Telephone numbers. Afr. Mat. 35, 17 (2024). https://doi.org/10.1007/s13370-023-01159-0. The authors purposely avoided the following , they must add these articles which are more related to their study.

7. Initial Coefficient Bounds for Bi-Univalent Functions Related to Gregory Coefficients Mathematics 2023, 11(13), 2857; https://doi.org/10.3390/math11132857

8. Korean J. Math.32(2024), No. 2, pp. 285 295https://dx.doi.org/10.11568/kjm.2024.32.2.285FABER POLYNOMIAL COEFFICIENT ESTIMATES FORANALYTIC BI-UNIVALENT FUNCTIONS ASSOCIATED WITH GREGORY COEFFICIENTS.

Reply to points 6-8:

Dear Respected Professor,

You are correct that the papers mentioned are closely related to our study. Unfortunately, we were unable to include them in the initial submission. However, we have incorporated these relevant studies into the revised version, as highlighted.

Thank you for bringing this to our attention.

Thank you again for your insightful feedback. We hope that this revised version will satisfy the requirements for publication.

Response to Reviewer 2

I would like to extend my sincerest gratitude to the editor and reviewers for their meticulous evaluation, insightful suggestions, and constructive comments, which significantly enhanced the quality and clarity of this article.

Reply to the Reviewer #2:

1. On page 1, line 5: remove the extra “can” from the following sentence” A can can be written as:”

2. On page 4, line 17: Replace “precise bound” with “sharp bounds”. We cannot write precise bounds.

3. On page 4: write “ and “ betwee the equations on line 22 and 23.

4. On page 5: Remove the period in Definition 1.

5. On page 6: In the last line of Lemma 2, it should be like this|𝑥|≤1,|𝑦|≤1.

6. On page 6: In the second line of Lemma 4, replace 0<𝐴<1 with 0<𝑇2<1

7. On page 7, line 8: Remove “period” and put the “ comma” after (3.9)-(3.12)

8. On page 11, line 25: “By the inequality (??) of Lemma 3, we have”. Equation number is missing.

9. On page 13, line 6: replace "v" with " using".

10. On page 14, line 2: remove the period

Reply: Dear Respected Professor,

We are pleased to report that we have thoroughly addressed points 1-10 in the revised version. Additionally, we have highlighted all revisions and incorporated new references closely related to our study.

11. The conclusion section lacks clarity on future directions. Please revise and expand the conclusions to include specific future research directions.

Reply: Respected Professor: We have incorporated future directions into the revised version. We believe this revision strengthens our manuscript and look forward to potential acceptance.

---

## [Decision Letter · Decision Letter 1]

27 Dec 2024

New class of symmetric starlike functions subordinate to the Generating function of Gregory coefficients

PONE-D-24-33833R1

Dear Dr. Khan,

We’re pleased to inform you that your manuscript has been judged scientifically suitable for publication and will be formally accepted for publication once it meets all outstanding technical requirements.

Kind regards,

Mohamed Kamel Riahi

Academic Editor

PLOS ONE

Additional Editor Comments (optional):

Reviewers' comments:

Reviewer's Responses to Questions

**Comments to the Author**

1. If the authors have adequately addressed your comments raised in a previous round of review and you feel that this manuscript is now acceptable for publication, you may indicate that here to bypass the “Comments to the Author” section, enter your conflict of interest statement in the “Confidential to Editor” section, and submit your "Accept" recommendation.

Reviewer #1: All comments have been addressed

Reviewer #2: All comments have been addressed

2. Is the manuscript technically sound, and do the data support the conclusions?

Reviewer #1: Yes

Reviewer #2: Yes

3. Has the statistical analysis been performed appropriately and rigorously? 

Reviewer #1: Yes

Reviewer #2: No

4. Have the authors made all data underlying the findings in their manuscript fully available?

Reviewer #1: Yes

Reviewer #2: Yes

5. Is the manuscript presented in an intelligible fashion and written in standard English?

Reviewer #1: Yes

Reviewer #2: Yes

6. Review Comments to the Author

Reviewer #1: The authors have thoroughly addressed all recommendations in the revised manuscript. This study presents original research findings, which have not been published elsewhere. The conclusions are well-supported by the data and presented appropriately. The article is written in standard English. Based on my evaluation, I recommend this article for publication in PLOS ONE.

Reviewer #2: The results are correctly formulated and could be understand easily;

The proofs are correct and clearly presented, and there are no any mistakes in the proofs.

Also authors updated all said corrections so I recommend for publications

7. PLOS authors have the option to publish the peer review history of their article (what does this mean?). If published, this will include your full peer review and any attached files.

Reviewer #1: No

Reviewer #2: **Yes: **Gangadharan Murugusundaramoorthy

---

## [Editor Report · Acceptance letter]

PONE-D-24-33833R1

PLOS ONE

Dear Dr. Khan,

I'm pleased to inform you that your manuscript has been deemed suitable for publication in PLOS ONE. Congratulations! Your manuscript is now being handed over to our production team.

Kind regards,

on behalf of

Dr. Mohamed Kamel Riahi

Academic Editor

PLOS ONE